# Variations in Persistence and Regenerative Zones in Coastal Forests Triggered by Sea Level Rise and Storms

**Sergio Fagherazzi \*, Giovanna Nordio, Keila Munz, Daniele Catucci and William S. Kearney** 

Department of Earth and Environment, Boston University, Boston, MA 02445, USA
\* Correspondence: sergio@bu.edu

**Abstract:** Retreat of coastal forests in relation to sea level rise has been widely documented. Recent work indicates that coastal forests on the Delmarva Peninsula, United States, can be differentiated into persistence and regenerative zones as a function of sea-level rise and storm events. In the lower persistence zone trees cannot regenerate because of frequent flooding and high soil salinity. This study aims to verify the existence of these zones using spectral remote sensing data, and determine whether the effect of large storm events that cause damage to these forests can be detected from satellite images. Spectral analysis confirms a significant difference in average Normalized Difference Vegetation Index (NDVI) and Normalized Difference Water Index (NDWI) values in the proposed persistence and regenerative zones. Both NDVI and NDWI indexes decrease after storms triggering a surge above 1.3 m with respect to the North American Vertical Datum of 1988 (NAVD88). NDWI values decrease more, suggesting that this index is better suited to detect the effect of hurricanes on coastal forests. In the regenerative zone, both NDVI and NDWI values recover three years after a storm, while in the persistence zone the NDVI and NDWI values keep decreasing, possibly due to sea level rise causing vegetation stress. As a result, the forest resilience to storms in the persistence zone is lower than in the regenerative zone. Our findings corroborate the ecological ratchet model of coastal forest disturbance.

**Keywords:** ecological ratchet; persistence niche; regenerative niche; salt marsh expansion; sea level rise; storm surges; hurricanes; coastal forest; NDVI; NDWI

## 1. Introduction

Retreat of coastal forests with rising sea levels has been documented [1–4]; however, this forest response can be influenced by a variety of circumstances. Storm surges, saltwater intrusion, flooding, and drought can also have a significant effect on these regions [5,6]. Even if these events do not kill trees outright, they can affect saplings that tend to be more sensitive to environmental stressors, thus reducing rates of overall forest regeneration.

In fact, salt and flooding can suppress tree regeneration before the death of mature trees [1,7], so that at lower elevations only mature trees are present. Critical factors controlling soil and groundwater salinity are the height and frequency of storm surges compared to the elevation of the forest soil [8]. Groundwater also plays a critical role, reducing salinity in areas where underground freshwater fluxes are high [9,10]. Marsh species cannot fully develop under the shade of trees [11], so that tree dieback is necessary to allow a complete transition from forest to salt marsh. Often trees are killed during intense storms, when temporary soil salinity is very high and soil saturation caused by rainfall and ocean waters increase the likelihood of windthrow [12]. As the magnitude of these events increases with changing climate, one would expect increased stress on these coastal ecosystems as well.

The low lying forests in the Delmarva Peninsula are often flooded. Vanderhoof et al. [13] used Radarsat-2 and Worldview-3 imagery integrated with LiDAR data to map flooding extent in wetland forests in Maryland and Delaware. Their analysis indicates that the hydrological connections between streams and low lying areas control flooding extent. Jin et al. [14] used Landsat time-series imagery to monitor inundation patterns in two watersheds of the Delmarva Peninsula. They found that tidal wetlands are, on average, more inundated than forest and scrub-shrub wetlands, with the latter prone to flooding due to rainfall. Huang et al. [15] used Landsat and airborne LiDAR data to measure inundation in a coastal forest in Maryland. Their analysis shows that the inundated area increases five times during wet years with respect to dry ones. Flooding regime and topographic elevation also affect forest structure in low lying areas. la Cecilia et al. [16] studied the forested floodplains of the Apalachicola River in Florida with Normalized Difference Vegetation Index (NDVI) data derived from Landsat images. They discovered that hardwood swamp has been partly replaced by bottomland hardwood forest in the last 30 years, predominantly near river banks, where the bottom elevations are higher.

The normalized difference water index (NDWI) has also been used to retrieve vegetation water content, and therefore, can monitor plant water stress [17–19]. Hamzeh et al. [20] compared NDVI and NDWI values derived from a Hyperion level 1B1 image to soil salinity levels measured in sugarcane fields. The good agreement of their results indicates that both NDVI and NDWI can be used to map salinity stress of crops. Similarly, Penuelas et al. [21] studied the effects of a soil salinity gradient on the spectral reflectance of barley measured in the field. Both NDVI and NDWI were found to be useful indexes to detect the response of barley to salinity. Justin George and Kumar [22] used NDWI to characterize soil salinity in an alluvial floodplain. Their analysis showed that areas with high soil salinity had very negative NDWI values, whereas normal soils had higher values, although negative, possibly due to presence of denser vegetation.

NDWI has already been used with success to classify and monitor wetland vegetation [23]. However, the link between NDWI, vegetation stress, and soil salinity might be more elusive than in carefully managed crop agriculture. This is because wetland soils may be an overwhelming source of moisture, especially when there are gaps in wetland vegetation. Here we will use NDWI to determine variations in forest characteristics driven by sea level rise and storms. The results will be discussed, taking in account both vegetation stress and canopy gaps.

## 2. Ecological Ratchet Model of Marsh Transgression in a Forest

A recent field study in the coastal pine forest on the Delmarva Peninsula, Virginia, USA, has proposed that the forest in this region can be divided into two distinctive zones [24]. The first zone, referred to as the persistence zone, is characterized by lower elevations, mature trees, but no new saplings. The size of this zone is related to the amount sea level rose since forest establishment [24]. The second zone, the regenerative zone, is characterized by higher elevations, mature trees, as well as saplings, indicating that the forest here is regenerating.

The ecological ratchet model of Kearney et al. [24] starts with the establishment of a forest stand, as it happened in the Delmarva Peninsula after the peak in deforestation at the beginning of the 20th century [25] (Figure 1A). As sea level rises and forest matures, the salinity of the soil increases, thus hindering forest regeneration at lower elevations (Figure 1B). Infrequent storms kill trees in the forest (Figure 1C), but only in the regeneration zone, the dead trees are replaced by saplings, so that the forest can fully recover (Figure 1D). By killing trees at the marsh boundary, storms also enable salt marsh plants to move into the dead forests, taking advantage of light availability. As a result the lower boundary of the persistence zone moves upland. On the other hand, sea level rise is responsible for the transgression of the upper boundary of the persistence zone in the regenerative zone, because edaphic conditions become less and less favorable to forest regeneration. Thus, in this model of marsh transgression, storms intermittently move the lower boundary of the regenerative zone (move the round gear of the ratchet), while sea level rise prevents forest recovery in the persistence zone (sea

level is the ratchet pawl that blocks the wheel, not allowing the reverse movement). Note that in this model the upper elevation of the persistence zone is determined, as a first approximation, by how much sea level rose since the establishment of the forest, so that in theory it could be possible to delineate the persistence zone on a topographic map, if the elevation of the landscape beneath the forest is known [24]. Moreover, since the persistence zone is characterized by a sparse tree canopy and the absence of saplings, which are typically very green, it should appear in remote sensing images, using for example the NDVI index.

These proposed zones are based on field data at two sites on the southeastern coast of the Delmarva Peninsula in the Eastern Shore of Virginia National Wildlife Refuge [24] (Figure 2). Field surveys found saplings only at elevations above 0.92 m with respect to NAVD88 at one site and above 1.37 m at the other site, suggesting that the boundary between these two zones varies slightly along the coast. The area between 0.92 and 1.37 m will be referred to as the transition zone. Forest below 0.92 m will be referred to as persistence zone and forest above 1.37 m will be referred to as regenerative zone. Note that current mean sea level is 0.146 m below NAVD88.

Since the defining characteristics of these zones are primarily demographic and not visually distinct in high resolution aerial imagery, with a typical resolution of 1–5 m per pixel, an attempt will be made to find corresponding physiological characteristics that will allow for the differentiation of these zones using remote sensing spectral data. Therefore, the primary purpose of this study is to determine if these proposed persistence and regenerative zones can be differentiated with remote sensing data, and if years with large storm events are distinguishable based on the spectral characteristics.

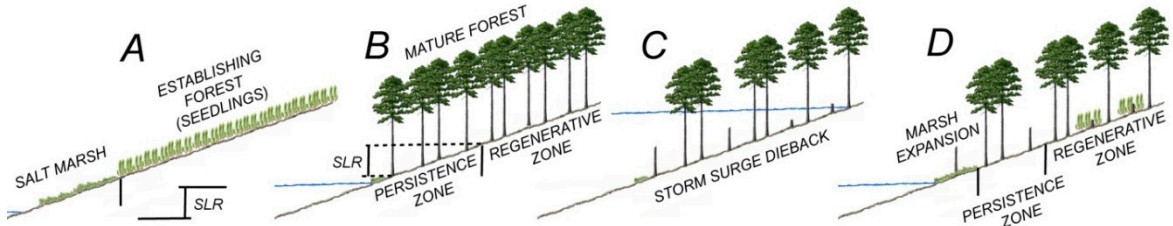

**Figure 1.** Ecological ratchet model of marsh transgression in a coastal forest [24]. After forest establishment (**A**) sea level rises, creating a persistence zone where mature trees survive under stress but cannot regenerate (**B**). A storm hits the forest triggering a dieback (**C**). Only in the regenerative zone seedlings can grow back and the forest recovers, while the marsh expands in the persistence zone (**D**). Note that the upper boundary of the persistence zone moves with sea level (press disturbance), while the lower boundary moves during storms (pulse disturbance).

## 3. Study Area

The Delmarva Peninsula is located along the Mid Atlantic coast of the USA and is one of the lowest lying areas in North America, with the highest point at 31 m above mean sea level [26] (Figure 2A). The rate of sea level rise in this area is very high, above 5 mm/yr [27]. A large swath of the original landscape of the region, dominated by forests and salt marshes, has been converted to agriculture in the last four hundred years [28] (see yellow areas in Figure 2B). However, given the low population density, many forested surfaces are still present, often bordering salt marshes [28]. Loblolly pine (*Pinus taeda*) is the dominant tree species of these forests [29], and it is often farmed as monoculture [28]. Sea level rose 7 cm in our study period from 2000 to 2019 [30], and 25 cm since the establishment of several forests in this area at the beginning of the 20th century [24].

A dendroclimatic study carried out in a low lying coastal forest in the Delmarva Peninsula indicates that the radial growth of loblolly pines is affected by disturbance due to storms [31]. A decline in radial growth was observed in the years of an extreme storm and lasted three years. After three years the radial growth resumed at a normal rate. The reduction in radial growth was correlated to the intensity of the storm. Forest resilience was assessed comparing radial growth in the three years after a

storm and in the three years before the storm. The forest was more resilient to storms occurring in winter (dormancy period) than in summer (growing season).

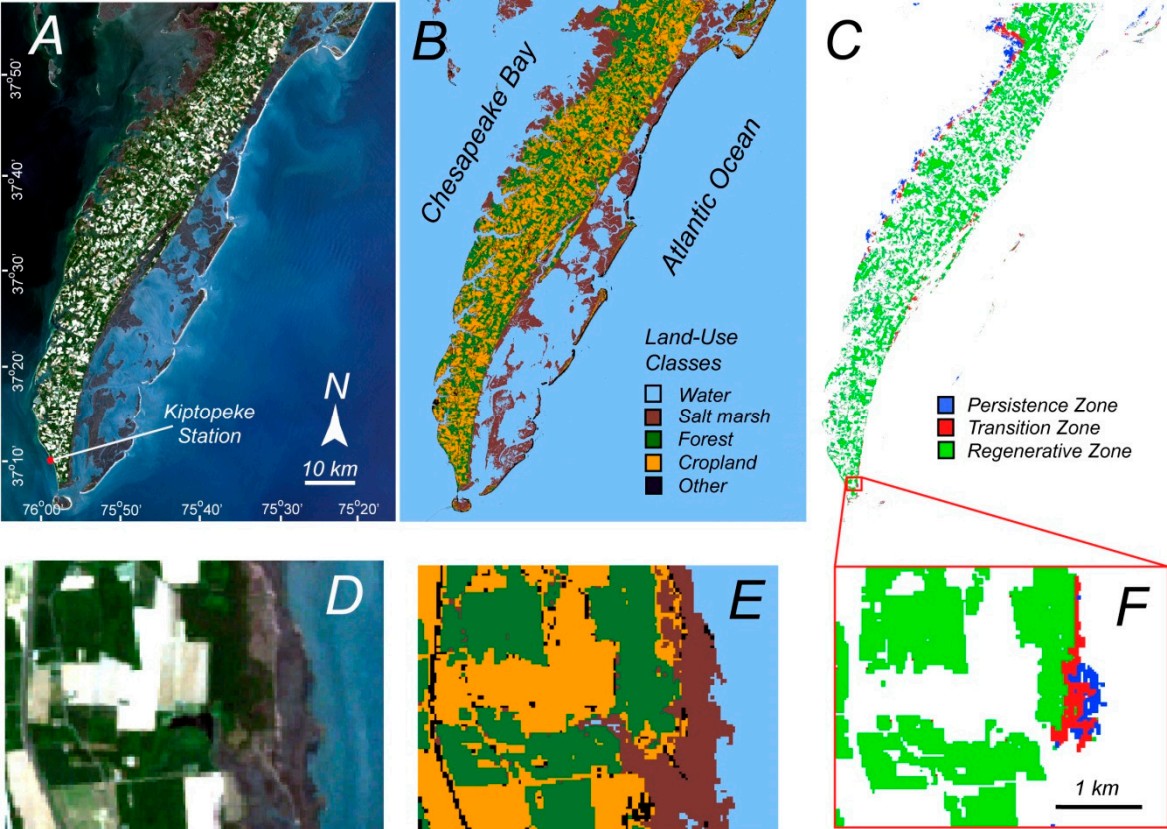

**Figure 2.** Delmarva peninsula along the East Coast of the USA. (**A**) Landsat 7 ETM+ C1 level 2 image taken on 4 May 2000; (**B**) land-use classes obtained with the Random Forest Classifier; (**C**) classification of persistence zone (forest below 0.92 m of elevation with respect NAVD88), regenerative zone (forest above 1.37 m) and transition zone (between 0.92 m and 1.37 m). (**D**–**F**) zoom-in of figures in the zone studied by Kearney et al. [24].

## 4. Methods

### 4.1. Forest Classification

To study the response of coastal forests to storms and sea-level rise we first need to identify the forested area in the Delmarva Peninsula. To this end we used a cloud-free Landsat 7 ETM+ C1 level 2 image taken on May 4 2000 (LE70140342000125EDC00) downloaded from the USGS Earth Explorer Database (Figure 2A). All bands were used for the classification. Herein we utilized a supervised classification based on polygons (quadrilaterals) with dimensions between 3 and 30 Landsat pixels. A total of 1140 training data (polygons) were selected as representative of 5 land cover classes (Table 1). Each polygon was chosen so that all its pixels belonged to a specific class by visual interpretation of the Landsat image. Larger polygons were used in homogenous areas, like open water. In particular, the class 'other' includes both sand and urban areas, since they are difficult to distinguish due to similar spectral signatures (Figure 3).

**Table 1.** Number of training data in each class.

| Class | Description | Training Data |
|---|---|---|
| Water | Ocean, Wetlands and Water basins inside | 99 |
| Marsh vegetation | Vegetation on marsh | 143 |
| Forest | Forested area at low and high elevation | 256 |
| Cropland | Agriculture settlements | 200 |
| Other | Sand and Urban area | 641 |

The supervised non-parametric learning algorithm adopted for the classification is the Random Forest Classifier (RFC) [32]. The RFC is considered a highly accurate and robust method, as it does not suffer from overfitting problems, because it involves a large number of decision trees participating in the process. Moreover, the RFC is easy to use because it operates with only two input parameters: The number of decision trees and the number of split variables at each node. These parameters are linked to one another, because the default number of split variables at the nodes is calculated as the square root of the number of total decision trees. Optimal results can be reached using an optimal number of decision trees and consequently of split variables minimizing the error. The RFC algorithm runs efficiently on large databases and it is nowadays a very popular classifier.

An accuracy assessment is needed to test the degree to which the produced classification of the image agrees with the reference classification [33]. Here we use stratified random samples with one stratum for each class, and the number of total sample size computed by the Cochran [34] equation:

$$n = \left( \frac{\sum (S_i W_i)}{S(P)} \right)^2 \approx \sum n_i \tag{1}$$

where $n_i$ is the number of samples (Landsat pixels) for the class (stratum) $i$, $W_i$ is the mapped proportion of area of class $i$, and $S_i$ is the standard error related to the stratum $i$, calculated as:

$$S_i = \sqrt{p_i(1-p_i)} \tag{2}$$

where $p_i$ is the proportion associated to the study question. We fix $p_i$ to 0.9 for the forest class, assuming that the forest class has an accuracy of 90%, and that only 10% of the forest area is misclassified. Since open water, urban areas, and marshes have a very different spectral signatures compared to forest, while the cropland spectral signature is similar (Figure 3), we assign the remaining 0.1 to the cropland class only, assigning 0 to water, urban areas, and salt marshes. The target standard error of the estimated overall accuracy $S(P)$ is fixed at 0.003, which is in the typical range 0.003–0.005 proposed by Olofsson [33]. We thus obtained a $n = 349.16$ which we approximate to 330 units (Landsat pixels). This sample size ensures that each stratum has at least 50 samples (Table 2). We assigned more samples to the forest and cropland classes, which cover large parts of the Delmarva Peninsula.

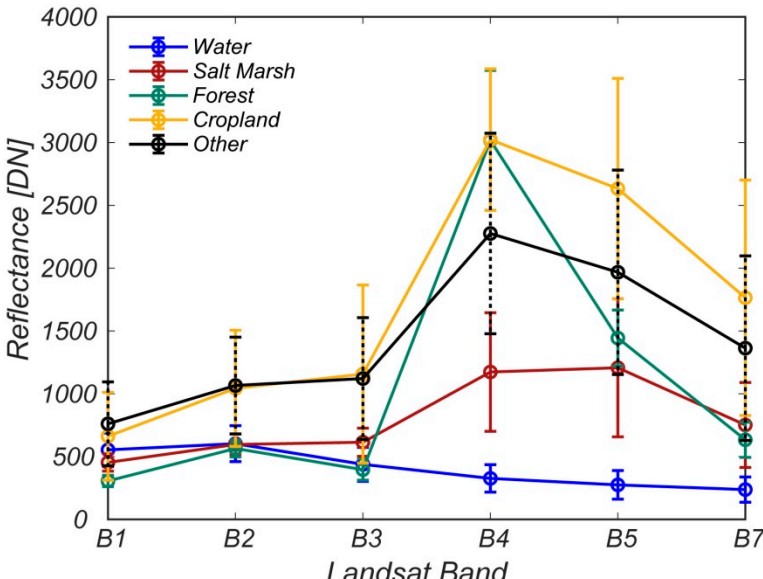

**Figure 3.** Multi-spectral surface reflectance curves for the five classes. Nodes represent the mean value and error bars represent standard deviations.

**Table 2.** Stratified random sample size and unit allocation for accuracy assessment.

|  | Water | Marsh Vegetation | Forest | Cropland | Other | Total |
|---|---|---|---|---|---|---|
| **number of pixels** | 8,720,340 | 823,993 | 1,067,549 | 1,173,132 | 206,330 | 11,991,344 |
| **number of pixels \*pixel's size (ha)** | 784,830.6 | 74,159.37 | 96,079.41 | 105,581.88 | 18,569.7 | 1,079,221 |
| $W_i$ | 0.727 | 0.069 | 0.089 | 0.098 | 0.017 | 1 |
| $p_i$ | 0 | 0 | 0.9 | 0.1 | 0 | |
| $S_i$ | 0 | 0 | 0.3 | 0.3 | 0 | |
| $S(P)$ | | | 0.003 | | | |
| $n$ | | | 349.16 | | | |
| $n_i$ | 50 | 50 | 100 | 80 | 50 | 330 |

　　The classification of the analyzed region is provided in Figure 2B, using the training data presented in Table 1. Through a first visual comparison between image and classified map, Random Forest Classifier seems to be able to recognize most of the features for each class. Sand and urban classes are the worst classified, being very often confused with cropland. This is due to the spectral signatures of features in the different classes that tend to be similar when croplands are drier, or when it is mixed with bare soil. From the classified image showed in Figure 2B, it is also possible to see how the cropland and urban/sand classes are characterized by same colors. The water class and the marsh class seem to be well represented, while the forest and cropland classes are sometimes confused with each other, due to similar spectral signatures when croplands are characterized by particularly green vegetation.

　　An error matrix can be obtained by comparing the classified map to aerial photographs taken by USGS on 30 December 2000. For the reference data in Table 1 we visually identified the land cover classes in the high resolution aerial photographs. The error matrix is expressed in term of counts (Table S1). Some omission and commission errors are present in all classes. In particular, for the class 'other,' in which sand and urban areas are combined, the range of error is high. On the other hand, 76 elements are correctly classified as cropland, and 88 elements are correctly classified as forest. The effective area estimate and the standard error associated to it are provided in Table S2. The overall accuracy, user's accuracy and producer's accuracy are indicated in Table 3.

**Table 3.** Accuracy table for forest classification method.

|  | **Water** | **Marsh Vegetation** | **Forest** | **Cropland** | **Sand/Urban** |
|---|---|---|---|---|---|
| Overall accuracy |  |  | 93.1% |  |  |
| User's accuracy | 98.0% | 80.0% | 88.0% | 80.1% | 39.0% |
| Producer's accuracy | 98.6% | 95.7% | 86.1% | 80.4% | 21.0% |

Table S2 provides a confidence interval for each class area. On the basis of the reference data, the area of the forest class is slightly underestimated of about 2%. On the other hand, Table 3 shows a good overall accuracy and a quite good user's and producer's accuracy for each class, except for the sand/urban class, where both accuracies are very low. In general, producer's accuracies (the accuracy from the point of view of the map maker) are better than user's accuracies (the accuracy from the point of view of a map user), due to the availability of more accurate reference data.

## 4.2. LiDAR Data and Storm Surges

A LiDAR Digital Elevation Model produced by Sanborne Geosystems in 2011 [35] with a spatial resolution of 3 m and a maximum vertical error of 20 cm was used to determine the boundary between the hypothesized locations of the persistence and regenerative zones around the peninsula, based on the one site assessed in detail by Kearney et al. [24]. Kearney et al. [24] measured a minimum, mean, and maximum elevation of saplings equal to 0.92, 1.24, and 1.37 m, respectively. These boundaries allow the delineation of three hypothetical forest zones on a map: a persistence zone below 0.92 m, a transition zone between 0.92 m and 1.37 m, and a regenerative zone above 1.37 m (Figure 2C).

We identified the most significant storm events during the 2000–2019 period, based on tidal data measured by NOAA with an acoustic water level instrument at the station 8632200, Kiptopeke, Virginia. Table 4 reports the maximum water levels referred to NADV88 for each storm. Note that water levels and tidal range slightly vary along the Atlantic and Chesapeake shorelines, but they are highly correlated [36]. As a result the highest storm surges occur across the entire region.

**Table 4.** Maximum storm surge at the Kiptopeke NOAA station (8632200), Virginia, for different storms.

| Events | Maximum Storm Surge (m above NAVD88) | Date of Storm Surge |
|---|---|---|
| Hurricane Isabel | 1.34 | 18 September 2003 |
| Hurricane Gaston | 0.938 | 13 November 2004 |
| Tropical Depression Ernesto | 1.191 | 7 October 2006 |
| Nor'Ida (Tropical Depression Ida and Nor'Easter) | 1.503 | 13 November 2009 |
| Hurricane Irene | 1.345 | 27 August 2011 |
| Hurricane Sandy | 1.476 | 29 October 2012 |
| Nor'easter | 0.913 | 9 December 2014 |
| Extratropical storm | 1.199 | 4 October 2015 |
| Hurricane Hermine | 1.079 | 3 September 2016 |
| Hurricane Florence | 1.05 | 9 September 2018 |

## 4.3. NDVI and NDWI Indices

NDVI and NDWI are two indices used to detect vegetation and its health. NDVI is probably the most used and the most well-known index for vegetation detection [37]. The general expression of NDVI is defined as:

$$NDVI = \frac{\rho_{NIR} - \rho_{RED}}{\rho_{NIR} + \rho_{RED}} \tag{3}$$

where $\rho_{NIR}$ and $\rho_{RED}$ are respectively the spectral reflectances in the near infrared and in the red regions. This index varies from −1 to 1, depending on vegetation conditions. NDVI correlates to physical properties of vegetation canopies, such as leaf area index, fractional vegetation cover, vegetation condition, and biomass [38]. NDVI is very sensitive to atmospheric effects and clouds, which contaminate measurements, soil moisture, and spectral effects. Moreover, NDVI is known to saturate when applied to forests having a leaf area index (LAI) of 3 or greater [17]. Because of these

limitations, Gao [17] proposed a new index, defined as NDWI or Normalized Difference Water Index, which is sensitive to changes in liquid content of vegetation canopies:

$$NDWI = \frac{\rho_{NIR} - \rho_{SWIR1}}{\rho_{NIR} + \rho_{SWIR1}} \tag{4}$$

where $\rho_{NIR}$ and $\rho_{SWIR1}$ are respectively the spectral reflectances in the near infrared and in the shortwave infrared regions. This index varies from −1 to 1, as the NDVI does. Absorption by vegetation liquid water is negligible in the NIR region and it is weak in the Short-Wave Infrared band 1 (SWIR1), providing information about plant water content when the canopy is sparse. The NDWI is less sensitive to atmospheric effects than NDVI and it should be considered complementary to NDVI, not its substitute. According to Gao [17], dry vegetation and green vegetation differ from each other in terms of NDWI, because dry vegetation generally reaches a peak in terms of spectral reflectance in the SWIR1 region, while green vegetation reaches its peak in terms of spectral reflectance in the NIR region. This leads to lower NDWI values for dry vegetation. NDWI values also increase when the soil is wet.

In a preliminary analysis, Landsat 8 scenes from summer (28 June 2017 and 31 August 2017), fall (3 November 2017), and winter (21 December 2017) were downloaded from the USGS Earth Explorer Database. Images from multiple seasons were selected to see if there was any significant seasonal difference between NDVI and NDWI. Bands one to seven were stacked in QGIS to create a single file for each date. The NDVI and NDWI were calculated for each image using the raster calculator in QGIS. Univariate raster statistics (mean and standard deviation) for NDVI and NDWI above and below each boundary were calculated in GRASS GIS. In this preliminary analysis, all pixels in each zone were used for only four images; in the following analyses, only the average value of NDVI and NDWI were extracted for each image, but then compared in all available images.

All the points in the persistence zone (below 0.92 m) and in the regenerative zone (above 1.37 m) were located in Google Earth Engine, and the NDVI and NDWI time series data were downloaded for each zone from 2000 to 2019. We use all available Landsat images in this period, masking pixels with cloud cover and cloud shadows with the Fmask algorithm proposed by Zhu and Woodcock [39]. The images are evenly distributed across seasons for all years. A moving average of 50 temporal points (Landsat images) was used to eliminate seasonal oscillations in NDVI and NDWI. Finally, a linear regression was utilized to compute the NDVI and NDWI trends between 2000 and 2019.

### 4.4. Resistance, Recovery and Resilience Indexes

Forest resilience to storm events was evaluated by using NDVI and NDWI to compute three metrics (resistance, recovery, and resilience indexes) as described in Lloret et al. [40] for the persistence and regenerative zones. The disturbance period (Dr) is defined as the year of the storm event plus the two following years; the pre-disturbance period (PreDr), and the post-disturbance period (PostDr) are 3 years before and after the disturbance period. If the storm occurs after October, as in the instance of Nor'Ida in 2009, we consider the following year as the first year of disturbance. To evaluate the resistance, recovery, and resilience indexes, we compute the average and standard deviation of the normalized NDVI and NDWI indexes in the three periods Dr, PreDr, and PostDr. The normalized NDVI index, $NDVI_n$, and normalized NDWI index, $NDWI_n$, are defined as:

$$NDVI_n = \frac{NDVI - NDVI_{min}}{NDVI_{max} - NDVI_{min}}; \; NDWI_n = \frac{NDWI - NDWI_{min}}{NDWI_{max} - NDWI_{min}} \tag{5}$$

The normalization of the indexes is deemed necessary because in some years the average NDWI is negative, indicating stressed vegetation or large canopy gaps. Resistance, defined as the inverse growth reduction during the disturbance, was estimated as the average $NDVI_n$ and $NDWI_n$ between Dr and PreDr. Recovery, the ability of trees to recover relative to the damage experienced during the disturbance, was estimated as the ratio of average $NDVI_n$ and $NDWI_n$ between PostDr and Dr.

Resilience, the ability of the trees to reach their pre-disturbance growth levels, was estimated as the ratio of average $NDVI_n$ and $NDWI_n$ between PostDr and PreDr.

We apply the resistance, recovery, and resilience indexes to all storms with a surge above 1.3 m, which flood both the regenerative and transition zones (Table 4). The first event was Hurricane Isabel that hit the area in the summer of 2003. For this event, the PreDr period is 2000, 2001, and 2002, the Dr period is 2003, 2004, and 2005 and the PostDr period is 2006, 2007, and 2008. Starting in 2009, a series of hurricanes and storms with a surge above 1.3 m occurred in rapid succession: Nor'Ida in 2009, Hurricane Irene in 2011, and Hurricane Sandy in 2012. As a consequence, for some of these storms, it is impossible to compute the PreDr period and the PostDr period. We therefore decided to lump these three storms in one event with a Dr lasting 5 years. Therefore, for the Nor'Ida-Irene-Sandy event, the PreDr encompasses 2007, 2008, and 2009, the Dr. period is 2010, 2011, 2012, 2013, and 2014, and the PostDr is across 2015, 2016, and 2017.

## 5. Results

The average NDVI as a function of day of the year displays the typical seasonal variability, with higher values in summer (around 0.8 and higher) and lower values in winter (around 0.6–0.75) (Figure 4).

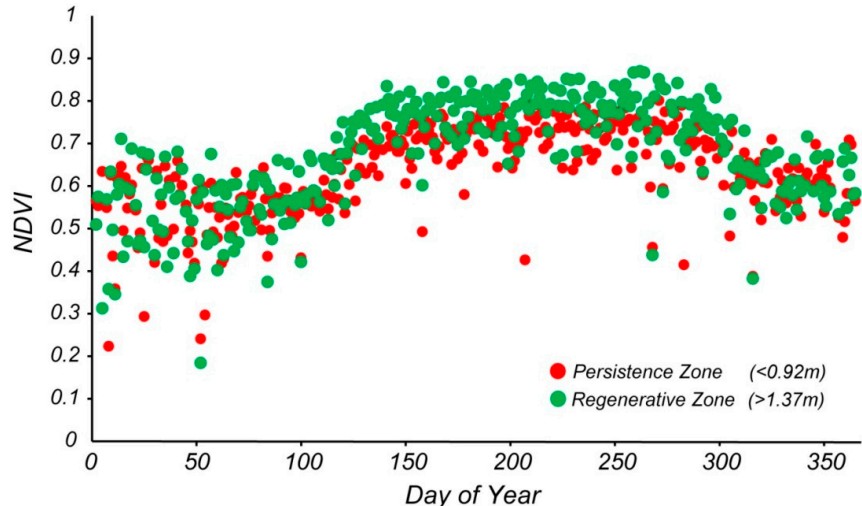

**Figure 4.** Average Normalized Difference Vegetation Index (NDVI) for all sample points for each day of the year, separated in persistence and regenerative zones.

More importantly, the NDVI in the persistence zone below 0.92 m is lower than the value in the regenerative zone above 1.37 m. To determine whether this difference is significant, and whether this difference varies if we change the elevation of the boundaries, we plotted the mean NDVI plus two standard deviations for different zones for four images taken in June, August, November and December of 2017 (Figure 5B). We focused on few selected images to account for the seasonal variability in NDVI within each zone, considering all pixels. Furthermore, we chose the images of 2017 because it was temporally distant from major storms, so the NDVI and NDWI values better represent the long-term forest conditions. We divided the forest based on the three boundary elevations of Kearney et al. [24]: Below and above the maximum sapling elevation of 1.37 m (Figure 5A), below and above the average sapling elevation of 1.24 m (Figure 5B), below and above the lower sapling elevation of 0.92 m (Figure 5C), along with the average NDVI for the zone below 0.92 m, and the zone above 1.37 m (Figure 5D). The Cohen's *d* effect size was computed to determine whether the difference in mean NDVI between these zones was large enough (Table 5). Figures 5A–C and 6A–C represent a sensitivity analysis of the regenerative and persistence zone with respect to elevation.

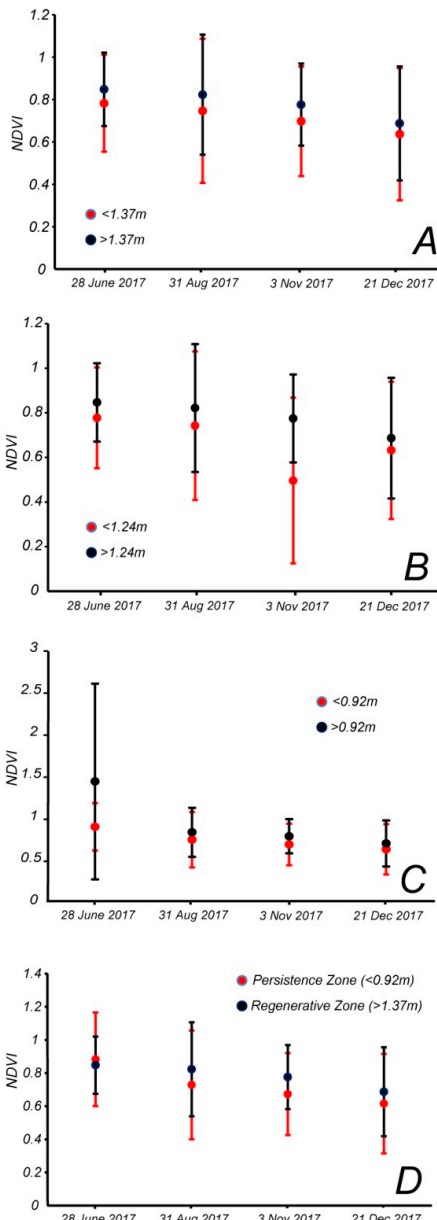

**Figure 5.** (**A**) Comparison of average NDVI for elevations below and above 1.37 m for four Landsat images in different seasons. (**B**) Comparison for elevations below and above 1.24 m. (**C**) Comparison for elevations below and above 0.93 m. (**D**) Average NDVI in the persistence zone (below 0.93 m) compared to average NDVI in regenerative zone (above 1.37 m). Bars represent two standard deviations.

**Table 5.** Cohen's *d* effect size for the difference between average NDVI computed below and above a specific elevation. The elevation separates the persistence and regenerative zones.

| Cohen's *d* Effect Size for NDVI | Boundary 0.92 m | Boundary 1.24 m | Boundary 1.37 m | Boundary 0.92 m–1.37 m |
| --- | --- | --- | --- | --- |
| 28 June 2017 | 0.98 | 0.74 | 0.71 | −0.35 |
| 31 Aug 2017 | 0.60 | 0.53 | 0.52 | 0.65 |
| 3 Nov 2017 | 0.94 | 2.36 | 0.75 | 1.02 |
| 21 Dec 2017 | 0.50 | 0.39 | 0.36 | 0.53 |

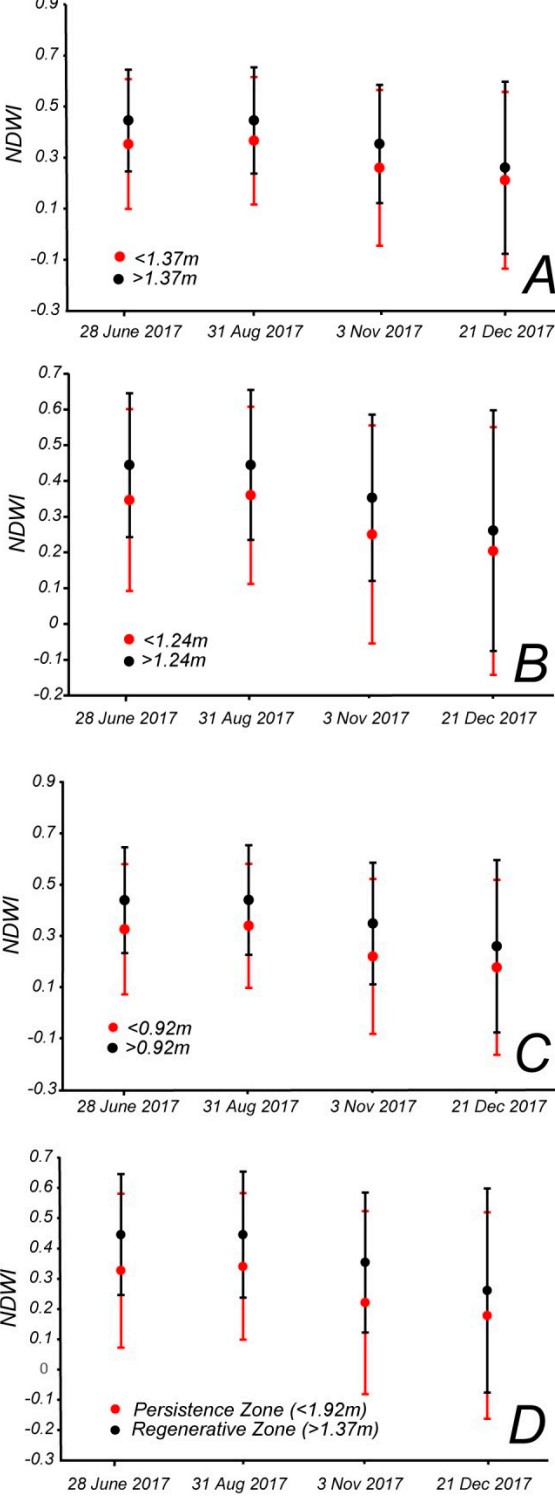

**Figure 6.** (**A**) Comparison of average NDWI for elevations below and above 1.37 m for four Landsat images in different seasons. (**B**) Comparison for elevations below and above 1.24 m. (**C**) Comparison for elevations below and above 0.93 m. (**D**) Average NDWI in the persistence zone (below 0.93 m) compared to average NDWI in the regenerative zone (above 1.37 m). Bars represent two standard deviations.

The average NDVI of the area above each elevation boundary is consistently higher than below the boundary, at all three elevations. The zones based on the 0.92 m boundary show a larger difference between average NDVI values, and the upper zone shows a larger drop moving from June to the end of August (Figure 5C). The zones based on the average sapling elevation of 1.24 m show a smaller

difference throughout the year, with the exception of a large drop in NDVI in the persistence zone in the fall (Figure 5B). The zones based on the 1.37 m upper boundary show a more consistent difference throughout the year (Figure 5A). The last plot, showing the difference between the average NDVI below 0.92 m (persistence zone) and above 1.37 m (regenerative zone), also appears to have a more consistent difference and steady decline throughout the year. However, in this plot, the NDVI of the persistence zone is larger than that of the regenerative zone in June (Figure 5D).

A similar analysis was carried out for the average NDWI and pairs of zones delineated by four possible boundaries: Below and above 1.37 m (Figure 6A), below and above 1.24 m (Figure 6B), below and above 0.92 m (Figure 6C), and below 0.92 m and above 1.37 m (Figure 6D). The Cohen's *d* effect size was also high between zones (Table 6). Overall, NDWI shows similar trends to NDVI. The NDWI in the upper zone is consistently higher than in the lower zone for all boundaries based on minimum, mean, and maximum sapling elevation. The difference between NDWI in each zone does not vary much, and the overall pattern throughout the year is pretty consistent for each elevation boundary (relatively similar throughout the summer and declining in the fall and winter).

**Table 6.** Cohen's *d* effect size for the difference between average NDWI computed below and above a specific elevation. The elevation separates the persistence and regenerative zones.

| Cohen's *d* Effect Size for NDWI | Boundary 0.92 m | Boundary 1.24 m | Boundary 1.37 m | Boundary 0.92 m–1.37 m |
|---|---|---|---|---|
| 28 June 2017 | 1.06 | 0.92 | 0.87 | 1.15 |
| 31 Aug 2017 | 0.93 | 0.78 | 0.73 | 0.99 |
| 3 Nov 2017 | 1.05 | 0.83 | 0.76 | 1.10 |
| 21 Dec 2017 | 0.49 | 0.34 | 0.29 | 0.49 |

For NDVI, the Cohen's *d* values are all positive and above 0.5 (medium effect size) in the June and August Landsat images, above 0.8 (large effect size) in November, and between 0.2 and 0.5 in December (small effect size) (Table 5). The difference is larger if we compare the zone below 0.92 m and the zone above 1.37 m, excluding the image of June, where the average NDVI above 1.37 is lower than the average NDVI below 0.92 m. For NDWI the effect size is always medium (above 0.5) or large (above 0.8) in June August and November, while it is small (between 0.2 and 0.5) in December (Table 6). Therefore the two zones seem better separated in summer and fall than in winter, when the NDVI and NDWI values decrease. The Cohen's *d* effect size is always higher if we compare the zone below 0.92 m to the zone above 1.37 m, excluding the NDVI values in June. The choice of the boundary elevation thus slightly affects the difference in average NDVI and NDWI values. Based on those results in the remaining analyses we separated the dataset in below 0.92 m (persistence zone) and above 1.37 m (regenerative zone). Those two zones yield the highest effect size. Forest with elevation between 0.92 m and 1.37 m (transition zone in Figure 2) was not assigned to either persistence or regenerative zone. In this way, we controlled for possible errors in LiDAR elevations that were around 20 cm.

The seasonal variability in NDVI during the study period is reported in Figure 7. We also indicate in the figure the ten storms that generated the highest storm surges since 2000 (Table 4). For each storm we report on the right y-axis the elevation of the storm surge. As expected, the NDVI values are higher in summer months for both zones (right-tailed *t*-test, $p < 0.05$) and lower in winter (left-tailed *t*-test, $p < 0.05$). Moreover, the moving average of the NDVI in the regenerative zone is always higher than the moving average in the persistence zone. In some winters, the NDVI values are reduced after a major storm (2004); in other years the reduction does not occur the year after a storm (2009). Summer peaks are also lower in some years (2004, 2011, and 2012) and they seem related to intense storms occurring in the fall of the year before. Additionally, the summer of 2003 displays a strong decrease in NDVI; this could be partly caused by hurricane Isabel, that struck on September 18. Note that in September the NDVI values are still very high (Figure 4) and they start declining in November.

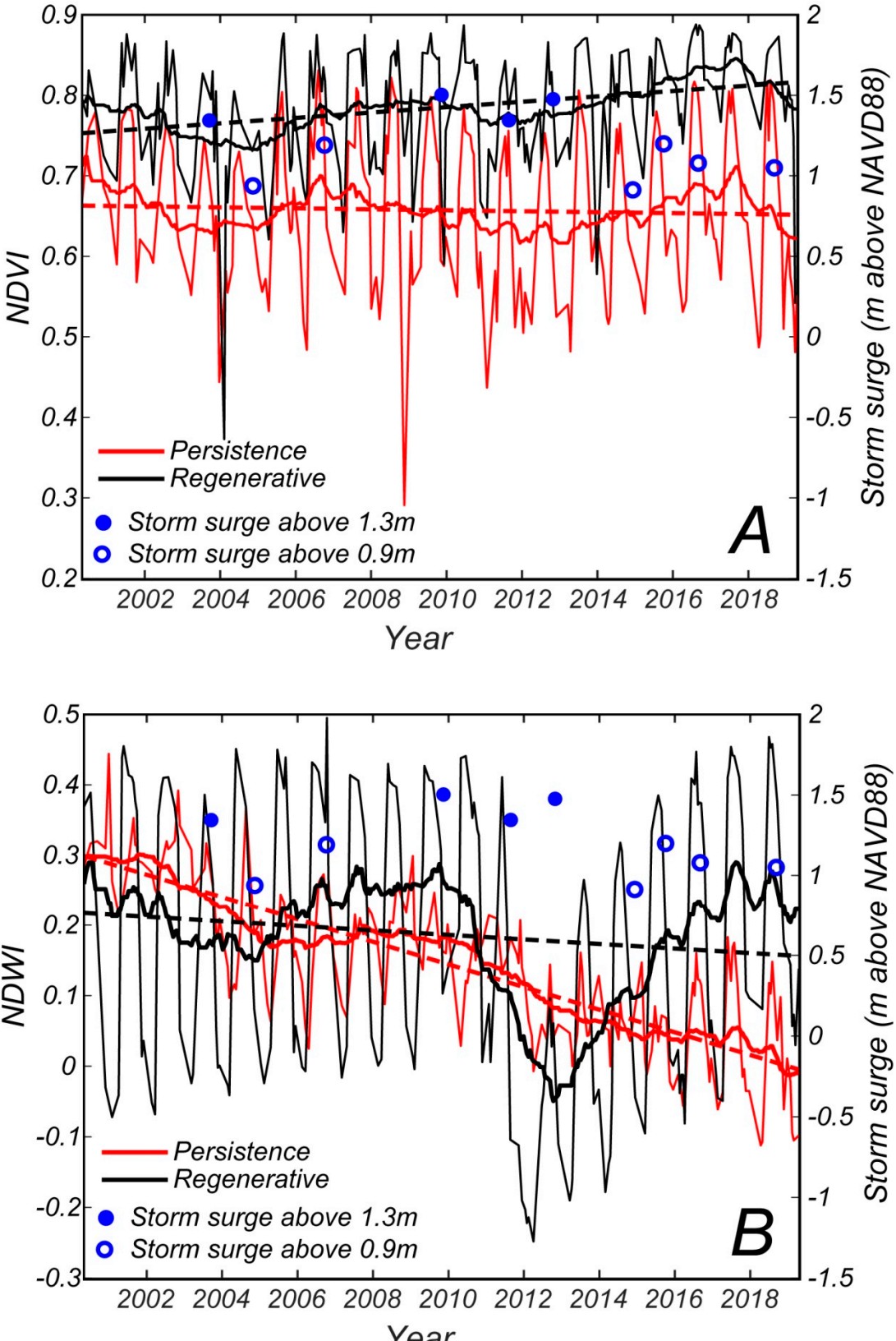

**Figure 7.** (**A**) Average NDVI values for persistence zone (below 0.92 m) and regenerative zone (above 1.37 m) for all images. (**B**) Average NDWI values for persistence zone (below 0.92 m) and regenerative zone (above 1.37 m) for all images. The thick lines are moving averages of 50 Landsat images. The dotted lines are linear interpolations. The highest storm surges during this period are also indicated as blue dots (maximum storm-surge elevation reported on the right y-axis).

The NDVI moving average, drawn as a thick line in Figure 7, represents more clearly the trend during the analyzed period. The detrended average NDVI in the regenerative zone is lower (negative) between July 2002 and June 2006, and between January 2010 and May 2015 (left-tailed *t*-test, $p < 0.05$, Figure 8), while it is higher (positive) between July 2006 and December 2009, and between June 2015 and December 2018 (right-tailed *t*-test, $p < 0.05$, Figure 8). The overall trend of NDVI is increasing from 2000 to 2019 in the regenerative zone ($r = 0.68$, $p < 0.05$, black dotted line in Figure 7A). The persistence forest moving average has a trend similar to the regenerative forest moving average—lower between June 2002 and August 2005, and between January 2009 and June 2015 (left-tailed *t*-test, $p < 0.05$, Figure 8); and higher in the periods September 2007 to December 2008 and July 2015 to December 2018 (right-tailed *t*-test, $p < 0.05$). The overall trend of NDVI in the persistence zone is a slightly decreasing one between 2000 and 2019 ($r = -0.16$, $p < 0.05$, red dotted line in Figure 7A). Inter-annual NDVI variations are larger for the persistence forest ($\sigma^2 = 0.010$) than for the regenerative one ($\sigma^2 = 0.005$).

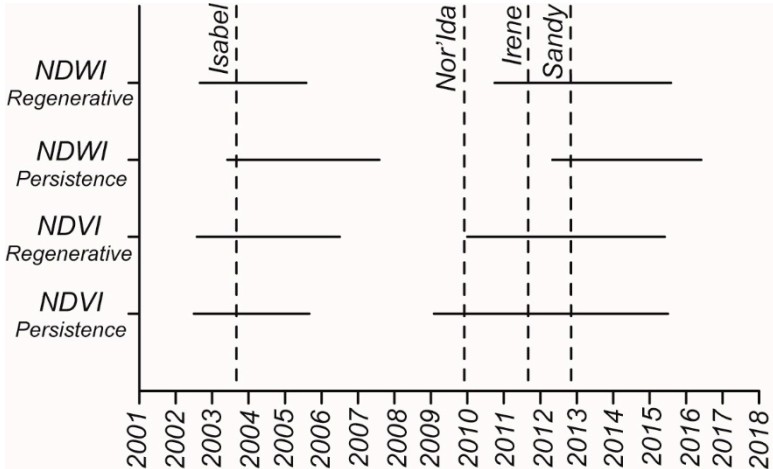

**Figure 8.** Periods with lower average NDVI and NDWI in the persistence and regenerative zones (negative anomaly of detrended moving average).

Figure 7B suggests that NDWI values for the regenerative and persistence forests are different. The regenerative forest achieves much higher NDWI values in summer (right-tailed *t*-test, $p < 0.05$). The NDWI moving average for this area, represented as a thick black line in Figure 7B, shows a quite small reduction during the August 2003 to July 2005 period, and a very large decrease during the September 2010 to July 2015 period (left-tailed *t*-test, $p < 0.05$, Figure 8). On the contrary, the August 2005 to October 2010 and August 2015 to December 2018 periods are characterized by a high detrended average NDWI (right-tailed *t*-test, $p < 0.05$). The overall NDWI trend for the regenerative forest displays a small decrease over the 2000 to 2019 period ($r = -0.22$ $p < 0.05$, black dotted line in Figure 7B). The persistence forest achieves a maximum NDWI value of about 0.35–0.4 in the 2000 to 2002 period, while in the following years the NDWI is lower, varying between 0.1 and 0.3 (left-tailed *t*-test, $p < 0.05$). Minimum NDWI values decrease too, varying from 0.25 during 2000 to 2003 period to $-0.1$ during 2016–2018 period. The NDWI moving average is decreasing, although it shows a limited recovery in the periods from 2004 to 2010 and from 2016 to 2018. The overall decrease in NDWI in the persistence zone is confirmed by the regression line (($r = -0.97$ $p < 0.05$, red dotted line in Figure 7B). Inter-annual NDWI variations are larger for the regenerative zone ($\sigma^2 = 0.041$) than for the persistence zone ($\sigma^2 = 0.005$).

Figure 9A suggests that the difference between NDVI in the regenerative and persistence forest increases over the 2000 to 2019 period; and linear regressions of the yearly averaged values confirm this trend ($r = 0.73$, $p < 0.05$). Overall, the NDVI difference almost doubles during the 2000 to 2019 period. Moreover, the difference between NDWI in the regenerative and persistence zones increases over the 2000 to 2019 period, as a linear regression of the yearly average values clearly indicates ($r = 0.62$,

*p* < 0.05). The maximum differences in NDWI occur in 2010, when NDWI in the persistence zone is quite low in comparison to the NDWI in regenerative zone (Figure 7B), and in 2012, when NDWI in regenerative zone achieves its minimum value over the 2000 to 2019 period (Figure 7B). Overall, the NDWI difference increases more than twice during the 2000 to 2019 period.

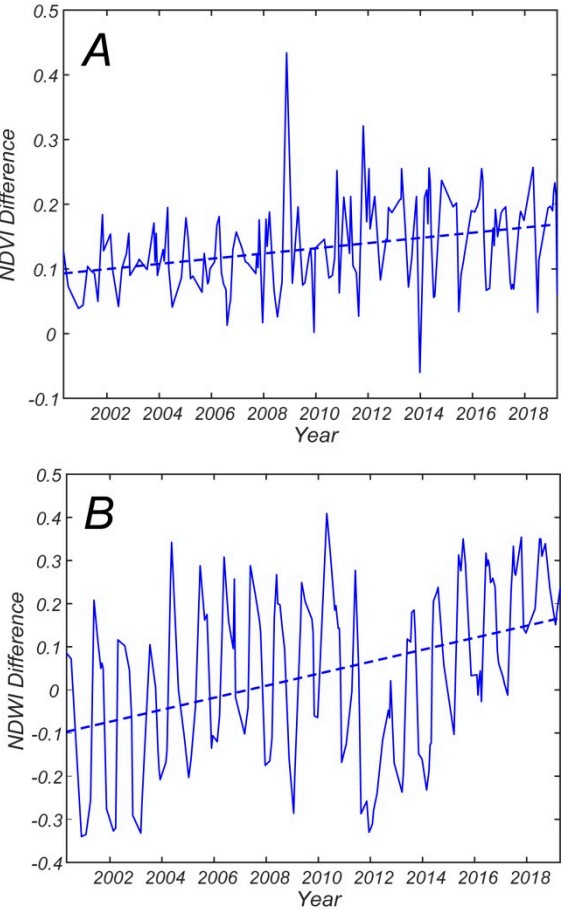

**Figure 9.** (**A**) Difference between average NDVI in regenerative and persistence forest. (**B**) Difference between average NDWI in regenerative and persistence forest.

The NDWI resistance index for hurricane Isabel in 2003 is similar in the persistence and regenerative zones (Figure 10A). The NDWI then recovers in the regenerative zone in the following three years but not in the persistence zone (Figure 10B), so that the resilience index is higher in the regenerative zone and equal to one (full recovery Figure 10C). In terms of NDVI, the resistance is slightly higher in the persistence zone (Figure 10D), but the recovery is much higher in the regenerative zone, although it displays a larger yearly variability (Figure 10E). As a result, the resilience index is slightly higher in the regenerative zone (Figure 10F).

NDWI resistance against the combined storms Nor'Ida, Irene, and Sandy is higher in the persistence zone (Figure 11A), while the recovery is much higher in the regenerative zone (Figure 11B), although with a larger temporal variability. As a result, the resilience is higher in the regenerative zone, and again, close to one (Figure 11C). In terms of NDVI, the resistance is higher in the regenerative zone (Figure 11D), while the recovery is higher in the persistence zone (Figure 11E). This results in a similar resilience index in the two zones (Figure 11F).

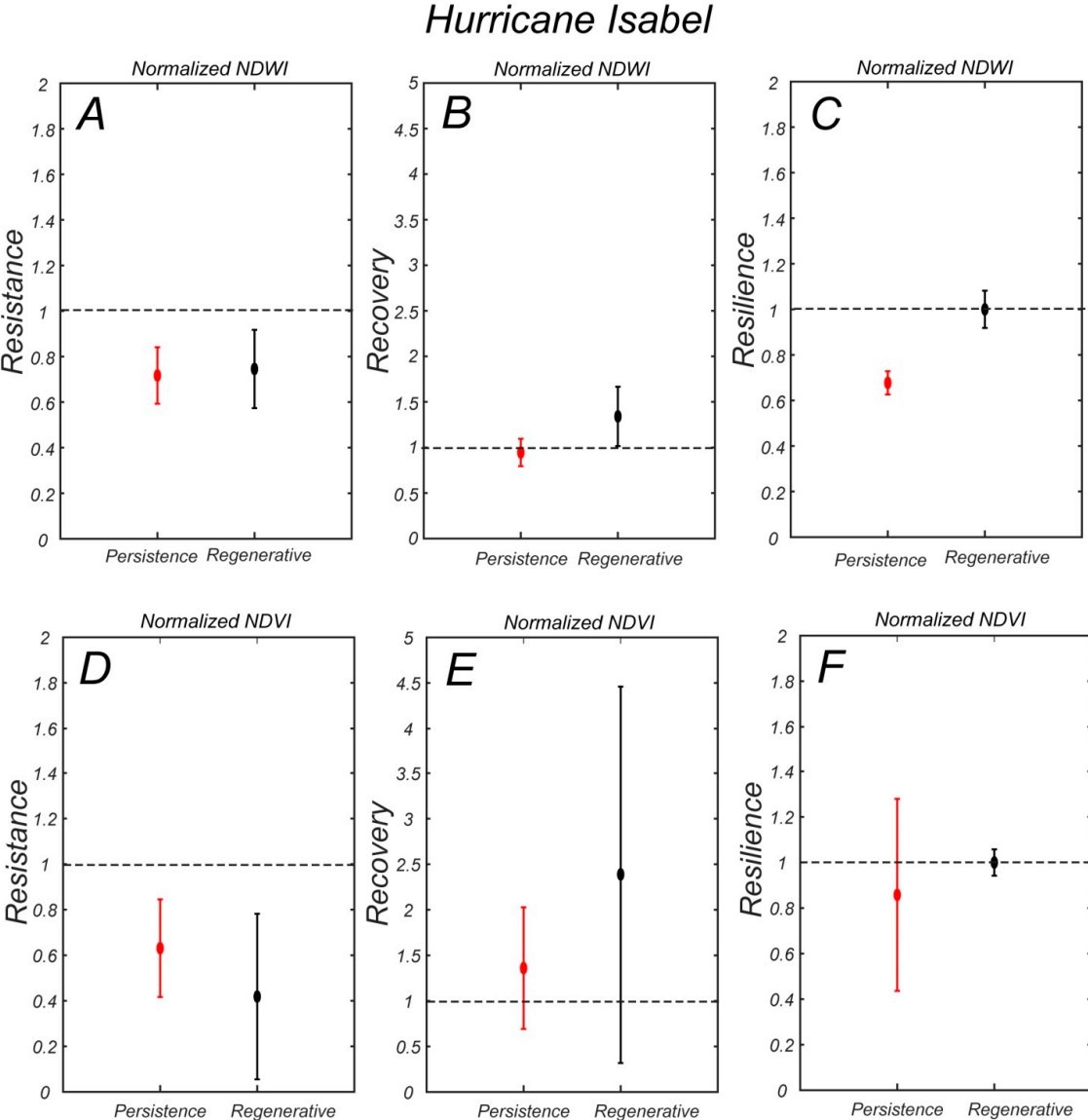

**Figure 10.** (**A**) Resistance, (**B**) recovery, and (**C**) resilience of persistence and regenerative zones computed with NDWI for Hurricane Isabel in 2003. (**D**) Resistance, (**E**) recovery, and (**F**) resilience computed with NDWI.

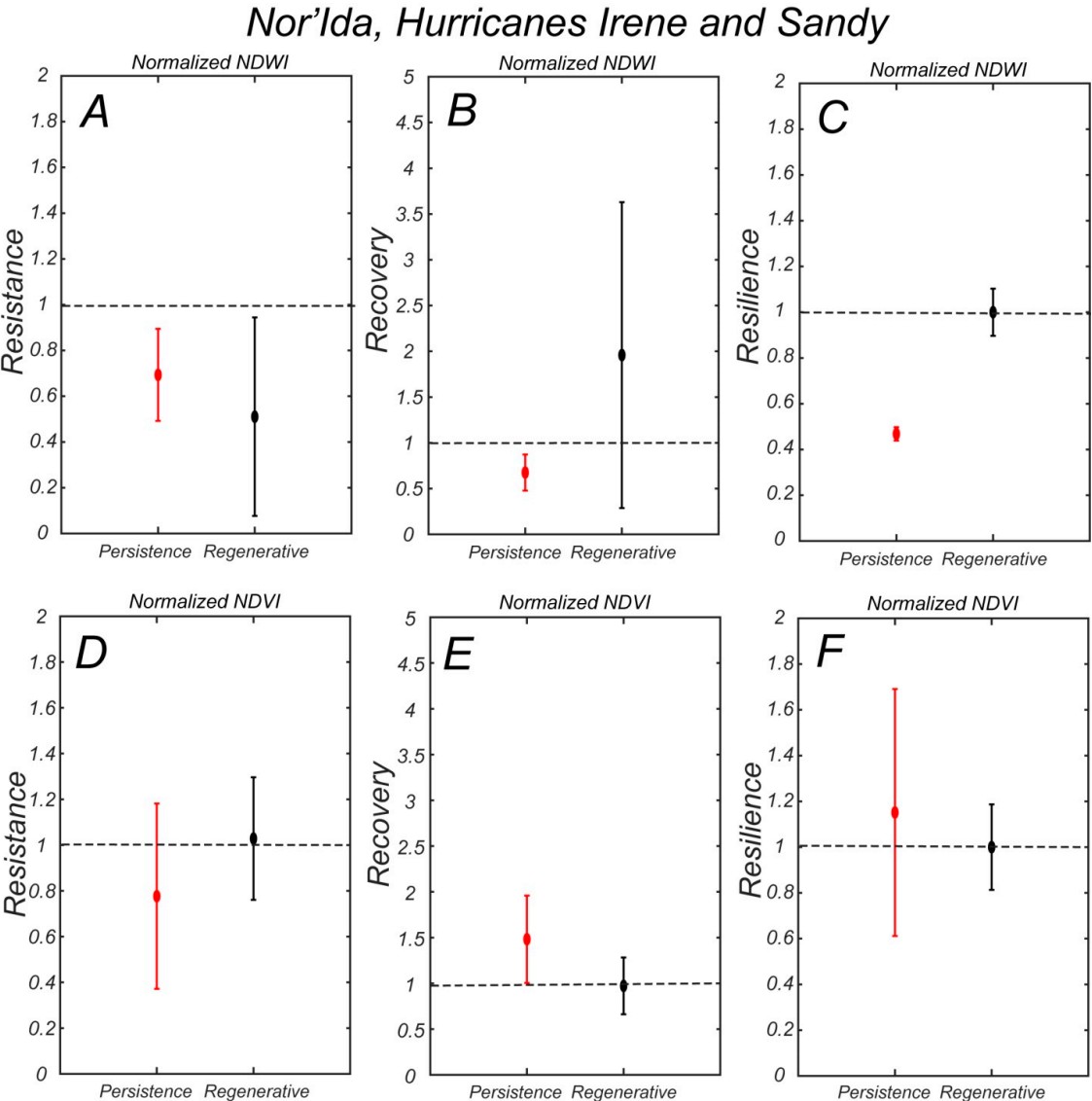

**Figure 11.** (**A**) Resistance, (**B**) recovery, and (**C**) resilience of persistence and regenerative zones computed with NDWI for the combination of Nor'easter of 2009, Hurricane Irene in 2011, and Hurricane Sandy in 2012. (**D**) Resistance, (**E**) recovery, and (**F**) resilience computed with NDWI.

## 6. Discussion

Both NDVI and NDWI indexes are consistently higher in the proposed regenerative zone than in the persistence zone, throughout the year. NDVI does show some more variation in how much it differs and when; however, this could be a result of the index capturing some of the broadleaf vegetation in the understory. This supports the idea that these two zones are distinctly different, both demographically and physiologically. Due to the scale of the area and the challenge of retrieving ground elevation below a forest canopy, the results of this analysis do not allow for clearer distinction between zones along the coast. In fact the margin of error of the LiDAR data is 20 cm, so it would be challenging to attribute the transition area between 0.92 and 1.37 m above NAVD88 to either the regenerative or persistence zone. Moving forward, this analysis should be expanded to see if this difference in average NDVI and NDWI can be used to more accurately represent this persistence–regenerative boundary at different points along the coast, instead of using a single elevation for the whole area.

The forest seems responsive to storms, particularly to those having a surge above 1.3 m with respect to NAVD88 (Figure 7). Four of such events occurred in the period 2000 to 2019. Both NDVI and

NDWI decreased after the storms (Figure 8); pronounced more so for NDWI (Figure 7). This index might be, therefore, more suited to study the effect of hurricanes on coastal forests.

One very interesting result is that the difference in NDVI and NDWI between the persistence and regenerative zones grows in time, possibly indicating the slow effect of sea level rise on the lower zone. An increase in salinity (in both soil and groundwater) would prevent the growth of very green saplings and understory vegetation, leading to a difference in NDVI. NDVI in the upper zone could also increase due to forest succession, with a canopy becoming denser and greener as the forest matures. However, forest succession would not explain the decrease in NDVI in the persistence zone or the decrease in NDWI in both zones. In fact, succession should also occur in the lower zone and NDWI should increase as the forest matures [41]. The increasing difference in NDVI and NDWI could also be driven by the transient response to the 2009–2012 storms, without representing a long term trend. This could be particularly true for the NDWI in the regenerative zone, which displays large oscillations. However, the increase is NDWI difference is mostly driven by a steady decrease in NDWI in the persistence zone, which is only slightly affected by the transient response (Figure 7).

In our study area, sea level rose only 7 cm from 2000 to 2019, but it rose 25 cm since some of these forests were established at the beginning of the 20th century [24]. Following the conceptual model of Kearney et al. [24], it is the overall increase in sea level since forest establishment that prevents the recruitment of new saplings. A 25 cm increase in sea level, likely accompanied by higher water tables and soil salinity, is therefore responsible for forest changes in the persistence zone. The decrease in NDVI and NDWI values in the persistence zone do not directly measure a rise in sea level, but forest mortality and the failure to recruit new saplings.

The NDWI index proposed by Gao [19] is affected by soil moisture only when the leaf area index is low [42]. So it is plausible that a large increase in canopy gaps would expose more soil, thus affecting the NDWI value. Four different processes triggered by sea level rise could lead to a change in the NDWI index: (i) Higher groundwater levels that increase soil moisture; (ii) more frequent flooding by storm surges; (iii) an increase in soil salinity, that stresses mature trees; and (iv) an increase in canopy gaps due to forest dieback, which exposes moist soils. All these processes lead to either vegetation water stress or a change in soil moisture, which are both potentially captured by the NDWI index. Therefore, NDWI seems very well suited to monitor the overall effect of sea level rise and storms on coastal forests.

An increase in soil moisture would increase NDWI values, whereas we detected a decrease in our study area. The NDWI values measured at our site are positive, ranging between 0.1 and 0.3 (Figure 7B). Areas with sparse vegetation and wet soils are characterized by negative NDWI values [22]. We, therefore, deduce that the vegetation canopy has a strong effect on the index, and that NDWI is not merely measuring soil wetness. This is further corroborated by high resolution aerial images, showing a dense tree canopy covering most of the persistence and regenerative areas. Flooding by storms increases soil wetness, but for few hours only. Since we do not detect negative spikes in NDWI after hurricanes, we conclude that storm flooding does not affect the NDWI index. Similarly, an increase in soil moisture due to higher groundwater levels does not seem to be captured by the NDWI index.

Instead, after large storm surges, we measured a decrease in NDWI that lasted 3–4 years. This long-term effect is likely linked to damage to the trees, with a subsequent slow-recovery. Moreover a decrease in NDWI values is also followed by a slight decrease in NDVI values (Figure 7). Because the NDVI index is mildly affected by soil moisture, we conclude that vegetation stress or damage is captured by NDWI. Note that both vegetation stress due to soil salinity, and tree damage that augment canopy gaps, can decrease NDWI after a storm. Both processes are reducing the health of the forest and can be ascribed to the combined effect of sea level rise and storm surges. More research is clearly needed to separate the effect of vegetation stress from the effect of canopy gaps on the NDWI index.

The NDWI and NDVI values almost fully recover in the regenerative zone after a storm, while the indexes do not recover in the persistence zone, permanently decreasing. This indicates that the ecological ratchet model might be valid. In the ratchet model an extreme event reduces the NDWI

values in both the persistence and regenerative zones (the gear of the ratchet), which cannot then allow a return to the original condition in the persistence zone because of sea level rise (the pawl of the ratchet, which is present only in the persistence zone).

When Hurricane Isabel hit in September 2003 the average NDVI and NDWI in both regenerative and persistence zones were relatively low, and they stayed low even in the following year (Figures 7 and 8). While the timing of hurricane Isabel does not correspond with the beginning of the lower NDVI and NDWI period, the hurricane may have contributed to keeping these values low in subsequent years. After hurricane Isabel, the vegetation indexes remained low for a period between 2 years (NDVI in persistence zone) and 4 years (NDWI in persistence zone). We notice a negative spike in NDVI in both the regenerative and persistence zones in the winter following the hurricane, and a reduction in NDWI in October 2003.

After Nor'Ida in 2009, the NDWI in the regenerative zone started decreasing and kept decreasing when Irene hit in 2011 and Sandy in 2012. After that, the NDWI grew for a period of 4–5 years. A similar trend, although more subtle, is also present in the NDWI of the persistence zone and the NDVI of both persistence and regenerative zones. We, therefore, suggest that forest trees in the regenerative zone were affected by the combination of the three storms, and recovered only 3–4 years after Sandy. In fact, periods of lower NDVI and NDWI ended between summer 2015 and summer 2016 (Figure 8). Periods of lower NDWI in the persistence zone are shifted with respect to period of lower NDVI and NDWI in the regenerative zone (Figure 8). This is probably due to the steep overall decrease in NDWI in the persistence zone, which affects the beginning of the low NDWI periods. These results are in agreement with the tree cores analyzed by Fernandes et al. [31] at a location within our study area. Fernandes et al. [31] discovered that radial growth of loblolly pines is suppressed in the three years after a flooding event triggered by a storm surge. This supports the hypothesis that NDWI is linked to vegetation stress in coastal pine forests, as was already proven in agricultural fields [20,21].

The regenerative zone of the forest has lower resistance in terms of NDWI, but a higher recovery and resilience. This likely indicates that trees in this area are affected by the flooding and the increase in soil salinity following a storm, but they are able to fully recover afterward, when edaphic conditions return to normal due to rainfalls. Saplings in this area can be killed by the storm, leading to a sharp reduction in NDVI and NDWI. This area is therefore affected by the pulse effect of storms. This is in agreement with the data collected by Kearney et al. [24] at a site within the study area. They measured a sharp increase in groundwater salinity after storm surges in the persistence zone; and a gradual decrease in salinity after rainfalls. On the contrary, surviving trees in the persistence zone seem to have a higher resistance in terms of NDWI, possibly because they are somehow accustomed to higher soil salinities. The lack of saplings also minimizes the consequences of a storm. However, both the recovery and overall resilience of the persistence zone is low, indicating that the vegetation here is slowly deteriorating. This is probably due to the press disturbance of sea level rise.

The NDVI resistance, recovery, and resilience indexes display a similar behavior for hurricane Isabel, with higher resistance in the persistence zone, and a higher recovery and resilience in the regenerative zone (Figure 10D–F). However, they are different from the combined Nor'Ida, Irene, and Sandy storms. For that 'event' the resistance of the regenerative zone is higher; therefore, reducing the recovery also. This is because NDVI was slowly increasing in the regenerative zone (Figure 7A), and that long-term trend mitigates the effect of the storms, particularly when considering a 5 year window.

The decrease in NDVI and NDWI starts one year before hurricane Isabel in 2003 (Figure 7), but it is magnified after the hurricanes. Other processes might, therefore, affect the forest canopy in this area; for instance, rainfall patterns, draughts, and temperature. However, as also discussed by Fernandes et al. [31], those effects are weaker compared to hurricanes.

The timing of hurricane arrival may also plays a role in forest disturbance, as corroborated by the dendroclimatic study of Fernandes et al. [31]. The growing season for these coastal forests extends from May to October, as indicated by the high NDVI values in Figure 4. If a large storm strikes during these months, the forest could be is instantly affected. The NDWI value of the regenerative zone dropped in

the month following hurricanes Isabel and Irene, which hit in September and August respectively, becoming significantly lower than the monthly average for the period 2000 to 2012. Hurricane Sandy, which occurred at the end of October, and the Nor'Ida storm, which occurred in November, did not trigger a sudden decrease in NDWI in the regenerative zone. However, the forest seems affected by those storms in the following years.

The difference in NDVI between persistence and regenerative zones might not be due only to absence of saplings. Seasonal NDVI data collected in stands of evergreen loblolly pines do not show a significant difference between winter and summer months [16], while our data indicate a strong seasonal variation (Figure 7). Different species of deciduous oaks can be found within the pine forests of the Delmarva Peninsula [28], lowering the NDVI values in winter. We also ascribe this strong NDVI seasonality to other understory vegetation species, and not only to pine saplings. Understory vegetation is particularly common in low lying coastal forests, where storm surges and wind kills trees, creating gaps in the canopy [28].

## 7. Conclusions

In the seminal paper of Kearney et al. [24], the coastal forests of the Delmarva Peninsula were separated into two zones based on elevation with respect to sea level: A persistence zone below 0.92 m with respect NAVD88, where mature trees can survive but are unable to regenerate because of elevated soil salinity and frequent flooding; and a regenerative zone, above 1.37 m, where trees can regenerate and saplings are present. Here we have used remote sensing (Landsat images) to test whether these two zones are different and how they respond to storm surges triggered by hurricanes and nor'easters. We have found that:

(1) The regenerative zone has statistically higher values of NDVI and NDWI, indicating that the forest is less healthy in the persistence zone.

(2) The difference in NDVI and NDWI values between the two zones is increasing with time, indicating that the health of the forest in the persistence zone is deteriorating.

(3) Both NDVI and NDWI values decrease after large storm surges; for instance, after Hurricane Isabel in 2003, Irene in 2011, and Sandy in 2012, and the Nor'Ida storm of 2009. The NDWI values decreased more than the NDVI values, suggesting that this index is better suited to determine the effect of storm surges on coastal forests.

(4) In the regenerative zone, the reduction in NDWI after a storm is larger than in the persistence zone. However, in the regenerative zone the NDWI index recovers to values similar to pre-storm conditions after three years from the storm, whereas in the persistence zone the NDWI keeps decreasing.

(5) The persistence forest is more resistant to storms than the regenerative forest in terms of NDWI, but the regenerative forest recovers faster. Overall, the resilience of the persistence forest is low, whereas the resilience of the regenerative forest is high.

Note that the regenerative and persistence zones presented herein are hypothetical, and based on field data from only one site [24]. The elevation of the boundaries of both zones may have varied along the peninsula. Future research will focus on the automatic detection of persistence and regenerative zones in coastal areas based on remote sensing images.

**Supplementary Materials:** The following are available online at http://www.mdpi.com/2072-4292/11/17/2019/s1, Table S1: Error Matrix in sample counts. Table S2: Total area for each column, standard error and variability interval.

**Author Contributions:** Conceptualization W.K., S.F., G.N., and K.M.; methodology W.K., G.N., and K.M.; formal analysis G.N., K.M., D.C., and S.F.; writing—original draft preparation S.F., G.N., and K.M.; writing—review and editing S.F.; funding acquisition, S.F.

**Funding:** This research was funded by the USA National Science Foundation award 1832221 (Virginia Coast Reserve VCR LTER) and 1637630 (Plum Island Ecosystems PIE LTER).

**Acknowledgments:** We would like to thank the Virginia Coast Reserve VCR LTER, the BU Center for Remote Sensing, and the Eastern Shore of Virginia National Wildlife Refuge for supporting this research.

**Conflicts of Interest:** The authors declare no conflict of interest.

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
