# Peer review of "Variations in Persistence and Regenerative Zones in Coastal Forests Triggered by Sea Level Rise and Storms"

_remotesensing, doi:10.3390/rs11172019_

Round 1
Reviewer 1 Report
The issues raised in my review have been sufficiently corrected in the revison, and I don't have further suggestions for the manuscript. I recommend it be accepted for publication.
Reviewer 2 Report
The authors has addressed satisfactorily all my major and minor concerns regarding the first version of the manuscript. I would suggest that the manuscript is in a good shape to be accepted for publication.
This manuscript is a resubmission of an earlier submission. The following is a list of the peer review reports and author responses from that submission.
Round 1
Reviewer 1 Report
This study utilized Landsat images to analyze the variations in the coastal forests of Delmarva Peninsula, and how they respond to the storm surges triggered by hurricanes. The manuscript is interesting to read, well organized and presented. However, I have a few comments/suggestions:
The title should be changed to something like "Detection of the variations in persistence and regenerative zones triggered by sea level rise in coastal forests using Landsat images". Sea level rise triggers variations, not the "zones".
Line 25, keywords are missing.
page 8, line 241, "resilience" -> "regenerative"
page 14, line 341, "Figure 9b" -> "Figure 6b"
page 16&17: The resistance, recovery, and resilience of persistence and regenerative zones of Hurricane Isabel (Fig. 8) and Nor’easter, Irene and Sandy (Fig. 9) are consistent in terms of NDWI, but inconsistent in terms of NDVI. In Figure 9, The resistance, recovery, and resilience of persistence and regenerative zones are opposite for NDWI and NDVI. Could you add some further explanations for those differences?
Reviewer 2 Report
Detection of persistence and regenerative zones triggered by sea level rise in coastal forests using Landsat images
General comments
This manuscript aims to verify the existence of a persistence and regenerative zone (as defined by the ratchet model of Kearny et al. 2019) on the coastal forest of Delmarva Peninsula, Maryland, USA by using Landsat imagery and NDVI and NDWI based indicators. The authors have analysed all available Landsat data for the period between 2000 to 2019, extracted the NDVI and NDWI indices for all forest points. Coastal forest points have been identified using a random forest classification algorithm. The time series has been clustered as belonging to regenerative and persistence zone based on elevation range. The authors have shown that there are statistically significant differences between the indicators on both zones. They have also shown how different indicators of resistance, recovery and resilience (as defined by Lloret et al. 2011) varies before and after storms observed during the study period.
The combined use of artificial intelligence (for land type classification) with satellite derived vegetation indicators (NDVI and NDWI) to verify a coastal forest response to sea level rise and storm surges is interesting for the research community and adequate for the journal scope. This reviewer has identified sections that need to be improved and numerous format and style errors that will need to be addressed before this manuscript can be accepted for publication.
Content wise, the main criticism of this paper is regarding conclusions not been fully supported by the evidences provided. The authors argued that the continuous decrease of the NDVI and NDWI values on the persistence zone are possible due to sea level rise but they do not present any indication of how much sea level has raised during the study period (2000 to 2019). An indication of past sea level rise at rates above 5mm/year (L118) is the only indication of the scale of sea level rise. Assuming that sea level has rose at this rate during the 20 years of the series analysed, will indicate that sea level has rose on the order of 10 cm and of similar order than the accuracy of the elevation model used to delineate the forest zones. No increase on soil salinity for the study period is reported. In this context, the conclusion that the vegetation indicators at the persistence zone is deteriorating because of sea level rise (L464) is not supported by the evidences provided.
Format wise, there are a number of sections that are simply missing (e.g. keywords, Author Contributions, Acknowledgements, Conflict of Interest) as well as numerous typos and sections that need improvement as described on the specific comments and typos below.
Specific comments
L70 First time that Delmarva Peninsula is mentioned = include state and country (e.g. Maryland, USA)
L78 confusing wording “While the forest matures sea level rises” …will it be better “As sea level rises and forest matures”
L98 first mention to NAVD88, please add height of this datum relative to present mean sea level.
L103 what resolution (e.g. pixel size) is high resolution?. Google Earth is not a measure of resolution
Typos
L29 replace “Williams et at.” by “Williams et al.” (replace t by l on al.)
L32 replace “Fagherazzi et al” by “Fagherazzi et al.” (point missing after “al” as short for alia)
L126 missing dot after “al.”
Equation (1) what is Wi?
L189 what are producers’ accuracies and user’s accuracies?
L199 replace Table 3 by Table 6?
L199 what equipment is used to measure the tidal level?
L281 in Table 8 heading replace “NDVI” by “NDWI”